# Organic Salts of Pharmaceutical Impurity *p*-Aminophenol

**DOI:** 10.3390/molecules25081910

**Published:** 2020-04-21

**Authors:** U. B. Rao Khandavilli, Leila Keshavarz, Eliška Skořepová, René R. E. Steendam, Patrick J. Frawley

**Affiliations:** 1Synthesis and Solid State Pharmaceutical Centre (SSPC), Bernal Institute, University of Limerick, Limerick, Ireland; Leila.Keshavarz@ul.ie (L.K.); renesteendam@gmail.com (R.R.E.S.); Patrick.Frawley@ul.ie (P.J.F.); 2PSC Biotech Limited, Blanchardstown, Dublin 15, Ireland; 3Department of Chemical Engineering, University of Chemistry and Technology Prague, Technická 3, 16628 Prague 6, Czech Republic; skorepova@fzu.cz; 4Institute of Physics ASCR, Na Slovance 2, 182 21 Praha 8, Czech Republic

**Keywords:** paracetamol, 4-aminophenol or *p*-aminophenol, salt, impurity removal, particle size

## Abstract

The presence of impurities can drastically affect the efficacy and safety of pharmaceutical entities. *p*-Aminophenol (PAP) is one of the main impurities of paracetamol (PA) that can potentially show toxic effects such as maternal toxicity and nephrotoxicity. The removal of PAP from PA is challenging and difficult to achieve through regular crystallization approaches. In this regard, we report four new salts of PAP with salicylic acid (SA), oxalic acid (OX), l-tartaric acid (TA), and (1*S*)-(+)-10-camphorsulfonic acid (CSA). All the PAP salts were analyzed using single-crystal X-ray diffraction, powder X-ray diffraction, infrared spectroscopy, differential scanning calorimetry, and thermogravimetric analysis. The presence of minute amounts of PAP in paracetamol solids gives a dark color to the product that was difficult to remove through crystallization. In our study, we found that the addition of small quantities of the aforementioned acids helps to remove PAP from PA during the filtration and washings. This shows that salt formation could be used to efficiently remove challenging impurities.

## 1. Introduction

From a pharmaceutical point of view, unwanted substances existing in the final formulation are regarded as impurities or pollutants, and these impurities, even in minute quantities, sometimes compromise the stability and safety of the drug substance [1,2]. The International Conference on Harmonization (ICH) has given guidelines in order to control the presence of impurities in the final active pharmaceutical ingredient (API) [3]. It is not always possible to remove or control the impurities during the synthesis of APIs, so companies usually try to minimize their presence after completion of the synthetic pathway [4]. However, it is not straightforward to eliminate the range of structurally related impurities from the final product; therefore, impurity management becomes an expensive process. Hence, pharmaceutical companies dedicate their research in order to find new cost-effective methodologies to remove or avoid impurities from the final product [5].

4-Aminophenol or *p*-aminophenol (PAP) is one of the major impurities in paracetamol (PA) and mesalazine (MZ), which originates from the synthesis and/or as a primary hydrolytic degradation product from storage conditions and that is difficult to remove [6,7]. PAP can show adverse effects such as nephrotoxicity and teratogenic toxicity and can cause methemoglobinemia even at the lower concentrations [8,9,10,11,12]. PAP degrades under certain pH conditions, which can cause a dark coloration to the product. In recent years, biological treatments have gained attention due to their capability to remove the impurities efficiently and cost effectively [13]. However, amino phenols can damage the microorganisms due to their toxicity that would limit the usage of biological treatments to remove amino phenolic impurities [14]. Drug regulatory agencies such as the Food and Drug Administration (FDA) and European Medicines Agency (EMEA) have restricted PAP impurity levels to 50 ppm (0.005% *w*/*w*) in a drug substance due to its high toxicology profile [15,16,17].

Salts and cocrystals are some of the well-known techniques to improve the physicochemical properties of pharmaceutical and bioactive compounds and can be used in the purification of APIs [18,19,20,21,22,23,24,25,26,27]. Larger p*K*a variations between the components (usually ΔpKa > 3.0) can lead to the formation of salts, and salts often show greater solubilities than the single individual components [6,28,29]. Based on this knowledge, we prepared four organic salts of PAP in an attempt to efficiently remove this impurity from PA.

Removal of PAP from PA through crystallization is challenging and cannot be achieved by regular cooling crystallization, to the best of the knowledge of the authors. This may be due to the similar physical–chemical properties of the impurity with respect to the API. To increase the physical–chemical differences between the impurity and API in an attempt to remove PAP from PA, four new salts of PAP with oxalic acid (OX), salicylic acid (SA), l-tartaric acid (TA), and (1*S*)-(+)-10-camphorsulfonic acid (CSA) have been synthesized (Figure 1). All the salts have been analyzed using infrared spectroscopy (IR), powder X-ray diffraction (PXRD), differential scanning calorimetry (DSC), thermogravimetric analysis (TGA), and single-crystal X-ray diffraction (SCXRD). Furthermore, we have attempted to understand the behavior of the product with and without traces of these salt formers during the crystallization process of paracetamol (PA) in the given solvent system and temperature conditions.

## 2. Results and Discussion

### 2.1. X-ray Crystallography

Single-crystal data for all the four salts were collected (please see Table 1 for parameters) at room temperature using a three-circle Bruker D8 Quest diffractometer with a sealed tube Mo anode, Kα radiation = 0.71073 Å.

### 2.2. Crystal Structure Analysis

The solved crystal structures of PAP salts have been compared with respect to the conformation (see Appendix A) and crystal packing of the PAP cation (Figure 2). The cation of PAP has a rigid aromatic core with hydroxyl and ammonium groups. The conformations of PAP in all of the newly described salts differ only in the relative positions of the hydrogen atoms on these groups evidenced by the different rotation about the C–O and C–N bonds.

The proton transfer has been in all salts confirmed by the electron density maps (for example, see Appendix A) that clearly show that there are three maxima around the PAP nitrogen atom corresponding to the ammonium group.

#### 2.2.1. PAP^+^OX^−^ Salt (2:1)

The PAP^+^OX^−^ salt crystallizes in the monoclinic crystallographic system with the space group *P* 2_1_/*n*. In the asymmetric unit, there is one cation of PAP and half of an oxalate anion (it lies in a special position on an inversion center). On average, there are four molecules of PAP and two molecules of OX in the unit cell.

In this structure, the molecules are arranged in hydrophobic and hydrophilic layers. The hydrophilic layers consist of the aromatic cores of PAP, which are arranged in face-to-face offset π–π stacking. The hydrophilic layers consist of the oxalate anions and the hydroxyl and ammonium groups of PAP, which all interact via H-bonding. In the H-bonding network, the oxalate anion acts as an acceptor for both the hydroxyl and ammonium groups of PAP.

#### 2.2.2. PAP^+^TA^−^ Salt (1:1)

The PAP^+^TA^−^ salt crystallizes in the monoclinic crystallographic system with the space group *P* 2_1_. In the asymmetric unit, there is one cation of PAP and one hydroxytartrate anion. There are two molecules of PAP and two molecules of TA in the unit cell.

In this structure, the molecules are also arranged into hydrophobic and hydrophilic layers. The hydrophilic layers consist of the aromatic cores of PAP, which are arranged in face-to-face *zig-zag* π–π stacking. The hydrophilic layers consist of the tartrate anions and the hydroxyl and ammonium groups of PAP, which all interact via H-bonding. In the H-bonding network, the tartrate anion acts as acceptors for both the hydroxyl and ammonium groups of PAP. Each tartrate anion forms H-bonds with five PAP molecules and also with four other tartrates.

#### 2.2.3. PAP^+^SA^−^ Salt (1:1)

The PAP^+^SA^−^ salt crystallizes in the triclinic crystallographic system with the space group *P*
$\bar{1}$. In the asymmetric unit, there is one cation of PAP and one salicylate anion. There are two molecules of PAP and two molecules of TA in the unit cell. In this structure, the molecules are arranged into layers, where molecules interact both by H-bonding and by aromatic face-to-face interactions of pairs of PAP cations and edge-to-face between SA and PAP. The layers are then connected to each other by aromatic interactions between the neighboring SA anions. In the H-bonding network, all three ammonium hydrogens of PAP form a charge-assisted H-bond with the carboxylate groups of the salicylate anions. The salicylate carboxyl also accepts H-bonds from the hydroxyl groups of both PAP and salicylate, as the latter bond is intramolecular.

#### 2.2.4. PAP^+^CSA^−^·H_2_O Salt (1:1:1)

The PAP^+^CSA^−^·H_2_O^−^ salt crystallizes in the monoclinic crystallographic system with the space group *C* 2. In the asymmetric unit, there are two cations of PAP, two camphor sulfonate anions, and two water molecules. Identical to those in the other salt structures, in this structure, the molecules are arranged into hydrophobic and hydrophilic layers, but here, the layers are more complex. The sulfonate groups of CSA form an H-bonded network with the water molecules and the ammonium and hydroxyl groups of PAP (layer ‘H’). There is an aromatic layer of the PAP benzene cores arranged into the herringbone pattern (layer ‘AR’) [6,29]. The aliphatic cycles of CSA form the last layer, ‘AL.’ In the crystal structure, the layers alternate in the …H-AR-H-AL-H-AR-H-AL-H… pattern.

### 2.3. Crystallization Experiments

#### 2.3.1. Crystallization of Paracetamol in the Absence of PAP

In benchmark experiment 1, which was carried out in the absence of PAP, colorless crystals of PA were obtained as shown in Figure 3. PXRD and IR of this sample matched with the commercial sample, and these crystals were used to compare to the crystals obtained from experiments 2, 3, 4, 5, and 6.

#### 2.3.2. Crystallization of Paracetamol in the Absence and Presence of PAP

The PAP impurity was introduced during the crystallization of PA in experiment 2, in order to understand its effect on the PA physicochemical properties. The presence of the PAP impurity was shown to significantly impact the color and purity of the final PA product. The presence of decomposed/polymerized PAP gave the dark coloration of the PA final product (as shown in Figure 4). However, we were unable to detect this impurity with our current analytical methods using UV and HPLC (see Appendix A). Most likely, the impurities are a complex mix of many aromatic compounds.

#### 2.3.3. Crystallization of PA in the Presence of PAP and a Counterion

In experiments 3, 4, 5, and 6, the same quantity of the PAP impurity was introduced to the PA crystallization batch, and an equimolar quantity of PAP salt former (either OX/CSA/TA/SA) at 80 °C has been introduced. Due to the expected high solubility of the resulting PAP salts, the impurity solubilized easily and could be easily removed by washing with the mother liquor and additional washing steps. The colorless PA products were harvested and illustrate that the products contained significantly less PAP-colored impurities (see Figure 5).

HPLC is a widely used method to detect the PAP in the literature and it is not easy to detect the impurities generated from the PAP [30]. We recorded the HPLC analysis for all the obtained PA samples and found that there are no traces of PAP in the samples (see Appendix A). He et al. reported that the degradation of PAP in the presence of O_3_ generates the intermediate products that could only be detected by gas chromatography coupled with mass spectrometry (GC/MS) [31]. Our testing methods (HPLC and UV) can reveal the presence of PAP or any other detectable impurities in the obtained PA samples from the above experiments. We could not quantify or detect the dark-color impurity in the obtained PA samples with our available resources now. Visual identification is the only way to confirm that the product is free of colored impurities in the experiments (see Figure 4 and Figure 5). However, we are confident that our future collaborative studies will work toward quantifying and understanding the toxicology profile of this color impurity. SEM and particle size analysis experiments were also carried out on the PA samples obtained from the above experiments. These results revealed that the obtained crystals from the experiments involving salt formation have uniform-sized/shaped crystals and were comparable to the crystals obtained from the initial experiment, which was carried out under the absence of PAP (see Appendix A).

## 3. Materials and Methods

### 3.1. Materials and General Methods

All organic solvents and samples were purchased from Sigma-Aldrich and used without any further purification. Milli-Q water was used as a solvent for all the experiments.

IR spectra have been collected on a PerkinElmer Spectrum 100 FT-IR Spectrometer fitted with a PerkinElmer Universal ATR Sampling Accessory.

DSC measurements of all salts were performed on a DSC 214 Polyma, NETZSCH instrument. Typically, 3–5 mg of sample was accurately weighed into a hermetically sealed aluminum pan and heated to 300 °C with a 10 °C·min^−1^ heating rate, under a nitrogen gas flow of 40 mL·min^−1^.

Thermograms of all salts were measured with a Perkin-Elmer TGA 4000 with a heating rate of 10 °C·min^−1^ under a nitrogen stream of 20 mL·min^−1^ to detect traces of solvents in the crystalline samples.

An EasyMax 102 Advanced Synthesis Workstation (Mettler Toledo) has been used to perform impurity removal crystallization experiments using the salt formation technique.

PXRD data were collected on an Empyrean diffractometer (PANalytical, Philips) using Cu Kα1,2 radiation (λ = 0.1541 nm) at room temperature operated at 40 kV and 40 mA. The samples were scanned over the range of 4–40° 2*θ* using a step size of 0.02° 2*θ* and a scan speed of 0.02° 2*θ*/s. Single-crystal data for all the salts were collected at ambient temperature using a three-circle Bruker D8 Quest with a sealed tube Mo anode, Kα radiation = 0.71073 Å, and a Photon 100 detector.

High-pressure liquid chromatography (HPLC) analysis was performed on all the paracetamol samples after the EasyMax spiking experiments. An Agilent 1260 Infinity quaternary LC system in combination with a ZORBAX Eclipse XDB-C18 column (4.6 mm × 150 mm, 3.5 μm) has been used to calculate the ratio of paracetamol to p-aminophenol in the solid and liquid samples. The mobile phase consisted of a 0.01 M sodium phosphate buffer of pH 3 and methanol in a 0.15/0.85 (*v*/*v*) ratio, respectively.

Scanning electron microscopy (SEM) images of all the crystals after the EasyMax experiments were taken using a JEOL Carryscope. The crystalline samples on the sample holders were coated with a thin layer of gold before analysis.

### 3.2. Synthesis of the PAP Salts

Salts were obtained by both solvent-drop grinding and crystallization methods. PAP was mixed with the corresponding acid in the equimolar ratio except for OX, which was prepared in a 2:1 PAP:OX ratio. Both solids were mixed in a mortar and ground for 20 min using drops of a 9:1 ratio of methanol and water solvent mixture. The resulting powder was used for further analysis. Single crystals could be obtained after dissolving PAP and the corresponding acid in 9 mL of methanol and 1 mL of water upon slight heating. After 3 days, good-quality crystals of salts were obtained (as shown in Table 2) through evaporative crystallization for the single-crystal analysis.

### 3.3. Impurity Spiking Experiments

Initially, 11 g of PA was dissolved in 50 mL of isopropyl alcohol (IPA) and water 20:80 (*v*/*v*). In a second experiment, 400 mg of PAP was added that also dissolved upon addition. The mixture was heated to 80 °C and stirred to get a clear solution using an overhead mixer at 500 rpm. For these two experiments, stirring continued at this temperature for 2 h. Then, the solution was cooled to 5 °C to induce crystallization and the slurry was filtered under vacuum using Whatman qualitative grade 1 filter paper. The filtered material was washed two times with 20 mL of 20:80 (*v*/*v*) water and IPA.

The filtered and dried PA obtained from the above experiments was tested by PXRD, ultraviolet spectroscopy (UV), HPLC, SEM, and laser diffraction.

## 4. Conclusions

Four novel salts of PAP were prepared and characterized. In all four salts, the molecules were arranged into hydrophobic and hydrophilic layers. Salt formation of PAP was used to remove the coloration of the product from PA through recrystallization of the API. We foresee that this approach can be used more generally to remove other impurities from other systems, provided that the solubility of the impurity salt is higher than that of the API. We could only identify the colored impurity visually; however, we are optimistic that our future studies will progress in the detection and quantification of this dark-colored impurity and reveal its nature.

## Figures and Tables

**Figure 1 molecules-25-01910-f001:**
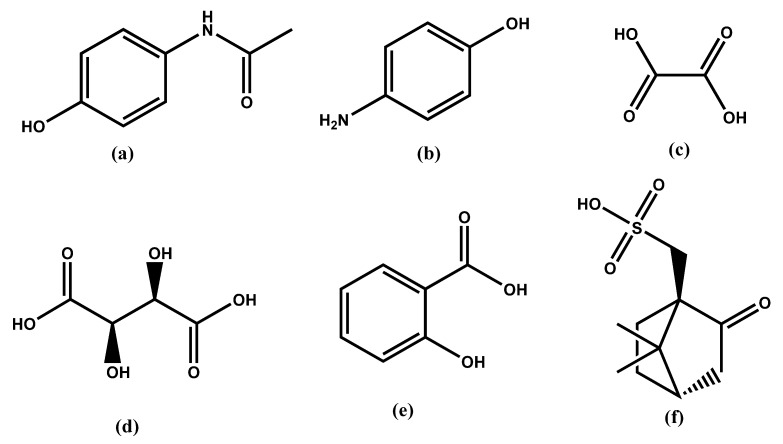
List of compounds used in this study: (**a**) paracetamol (PA), (**b**) *p*-aminophenol (PAP), (**c**) oxalic acid (OX), (**d**) l-tartaric acid (TA), (**e**) salicylic acid (SA), and (**f**) camphorsulfonic acid (CSA).

**Figure 2 molecules-25-01910-f002:**
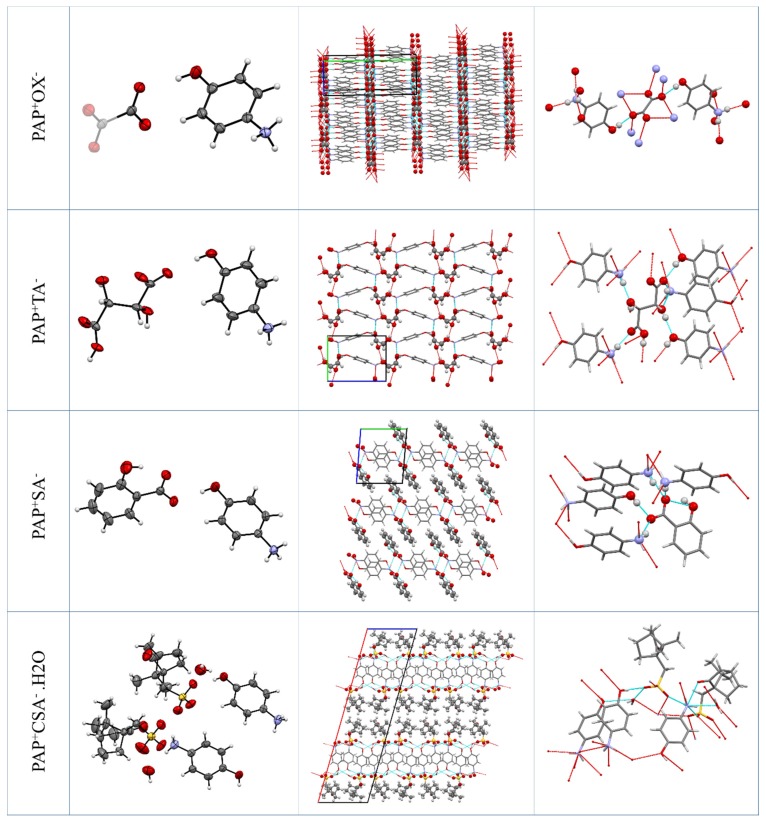
The crystal structures of the salts. Left–asymmetric unit with the thermal ellipsoids. Middle–molecular packing (anion highlighted). Right–details of H-bonding.

**Figure 3 molecules-25-01910-f003:**
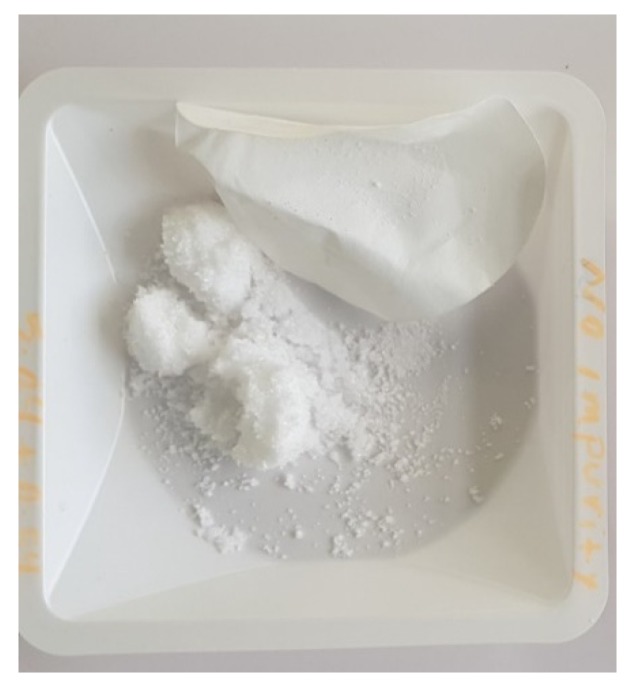
Paracetamol batch crystallized in the absence of the PAP impurity.

**Figure 4 molecules-25-01910-f004:**
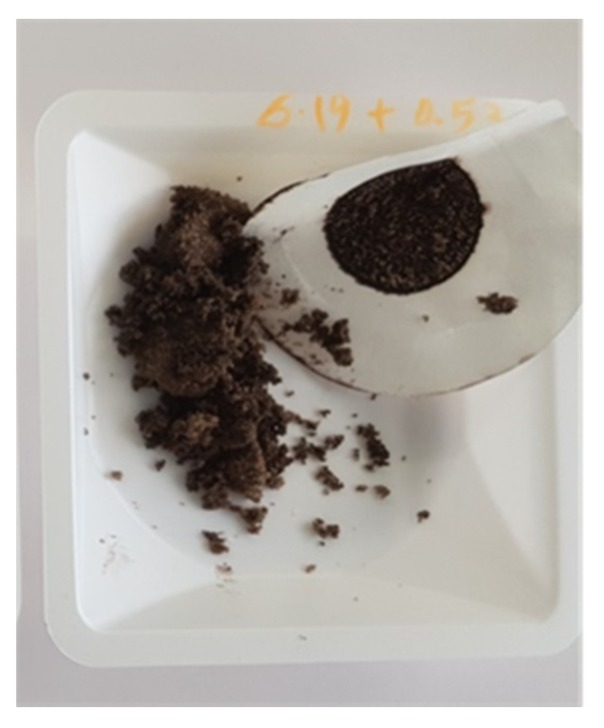
PA batch crystallized in the presence of the PAP impurity.

**Figure 5 molecules-25-01910-f005:**
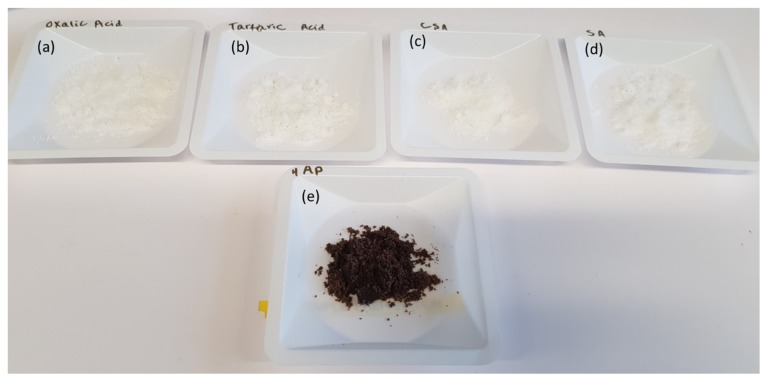
Crystallization experiments: (**a**) PA+PAP+OX, (**b**) PA+PAP+TA, (**c**) PA+PAP+CSA, (**d**) PA+PAP+SA, and (**e**) PA+PAP.

**Table 1 molecules-25-01910-t001:** Crystallographic parameters.

Compound	PAP^+^SA^−^	PAP^+^OX^−^	PAP^+^CSA^−^·H_2_O	PAP^+^TA^−^
Formula	C_13_H_13_NO_4_	C_7_H_8_NO_3_	C_16_H_25_NO_7_S	C_10_H_14_NO_7_
*MW*, g/mol	247.25	154.14	359.43	259.21
Crystal system	triclinic	monoclinic	monoclinic	monoclinic
Space group, *Z*	*P*$\bar{1}$, 2	*P* 2_1_/*n*, 4	*C*2, 4	*P* 2_1_, 2
*a*, Å	6.798(7)	6.6471(5)	42.654(2)	7.206(6)
*b*, Å	8.596(1)	17.357(7)	7.4090(3)	8.043(7)
*c*, Å	10.256(1)	6.768(5)	11.765(5)	10.53 (7)
*α*, °	93.756(4)	90	90	90
*β*, °	99.530(4)	117.233(5)	97.2740(1)	98.361(8)
*γ*, °	94.663(4)	90	90	90
*V*, Å^3^	587.2(6)	694.2(3)	3575.7(3)	603.9(1)
*D**c* gcm^−3^	1.398	1.475	1.335	1.426
μ, mm^−1^	0.105	0.117	0.212	0.123
2*θ* range, °	3.0–26.38	3.56–26.36	2.79–26.39	2.86–26.85
*T*, K	299	282	304	299
Total ref.	14,425	7997	23,035	9330
Unique ref.	2398	1414	7278	2480
Obs. ref., *I* > 2σ(*I*)	2046	1042	5649	2347
# Parameters	178	117	464	191
*R*_1_ [*I* > 2*σ*(*I*)]	0.0393	0.0450	0.0474	0.0727
*wR* _2_	0.0967	0.1003	0.1021	0.1601
*S*	0.9792	0.9748	0.9847	0.9779

**Table 2 molecules-25-01910-t002:** List of salts and their ratios.

Acid	Product (Ratio in the Bracket)
Oxalic acid (OX)	PAP^+^OX^−^ Salt (2:1)
l-tartaric acid (TA)	PAP^+^TA^−^ Salt (1:1)
*p*-Salicylic acid (SA)	PAP^+^SA^−^ Salt (1:1)
(1*S*)-(+)-10-Camphorsulfonic acid (CSA)	PAP^+^CSA^−^·H_2_O Salt hydrate(1:1:1)

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
