# Peer review of "Organic Salts of Pharmaceutical Impurity p-Aminophenol"

_molecules, 2020, doi:10.3390/molecules25081910_

Round 1

Reviewer 1 Report

Abstract

Melasalazine wasn't the topic of the paper and it is not necessary to be mention in the abstract.

Please explain all abbreviations at first appearance. Use abbreviations in figure descriptions.

Page 3, lines 114-119

It isn't necessary to repeat the same procedure for the experiment 2 which is same as the experiment 1. It would be enough to emphasize that was same as in experiment 1 with addition of a certain amount of impurities.

Page 8, lines 199-200

Differences in the purity between pure PA crystals and crystals obtained from the solution with dissolved PAP weren't proved with the methods used in the experimental work like authors write down in the next sentence. In my opinion it has to be proved with some other methods (eg. XRD or FTIR)

Page 9, lines 217-218

Please, explain obtained results.

Page 9, line 220

In my opinion statement about polymerised product should be proved. Would a higher amount of impurity indicate their presence in the used analytical methods or apply another analytical method to prove it? Furthermore, can you prove that the salts were obtained in synthesis, not the cocrystals. Please explain it.

Literature

Year in literature 4 is written in cursive instead bold, while volume is bold!

Author Response

We really thank for your comments and suggestion on our manuscript please find the attachment with the clarification and the changes we made. 

Reviewer 2 Report

This manuscript was a revised version from a previous one entitled “Removing 4-Aminophenol Impurity from 2 Paracetamol Using Salt Formation Approach”. It was now reframed around the report of four organic salts of p-aminophenol (PAP) which is pharmaceutical impurity of paracetamol drug. These PAP salts were characterized through single-crystal X-ray diffraction, powder X-ray diffraction, IR, DSC and TGA. The application of PAP salts employed as “impurity removal” was evaluated.

However, in my opinion, the manuscript still needs significant improvements in the report of crystal structures, crystal packing, hydrogen-bonding motifs, PAP conformation, etc.. This is necessary and important now because the report of salts’ structure becoming priority.

Moreover, the current writing form regarding to the report of crystallography and structure structures is far from reaching a scientific sound. In the current form, the crystallography data and structures are merely listed without further discussions in each section.

Other comments to be resolved:

#1. As I mentioned in the first round reviewing, the acknowledge to impurity purification using co-crystal approach must be made in the introduction section. At least, the following reference by Myerson et al. (CrystEngComm, 2012,14, 2386-2388) should be referred, given the similarities between co-crystals and salts.

#2. In table 2, too much significant digitals.

#3. It is unclear why packing similarity among PAP salts is necessary. From PXRD and crystallography data, there appears no evidence to suggest the presence of any similarities. In fact, figure 3 indeed does not show much similarity. All the values are greater than 10 (smaller than 5, considered to be similar).

Author Response

We appreciate your time to comment and suggestions on our manuscript please find the attached form with the answers/clarifications. 

Reviewer 3 Report

In my opinion, the revised manuscript does not answer to the main questions raised from the previous revising process. Namely:

  • The colour change itself does not represent a rigorous way to report an analytical data. Even if absolute quantification is not possible at least a relative value of the reomoved AP should be indicated (i.e. % of AP removed with respect to the starting amount of PA)
  • The HPLC data are still not clear. The figures reported in the SI are all similar and no comment has been done on these data nor in the text or in the SI. Indeed, this tool could be crucial to clarify the previous point on the PA removal. The authors should better exploit the hplc technique to support their results

In conclusion, I think that in this form the manuscritp does not meet the scope of the journal and I confirm that, unless the authors solve the mentioned issues, this manuscript would be more appropriate for a specialized journal on solid state characterization.

Author Response

We appreciate your time to comment and suggestions on our manuscript. Please find the attached form with the answers/clarifications. 

Round 2

Reviewer 1 Report

Please recheck the number of decimal places and fonts in Table 2. The usage of abbreviations in the abstract can be acceptable or not. It depends on journal's author guideline. Although, the authors use the abbreviations in the abstract (which is correct), some commonly understood and used abbreviations are thrown out (eg. IR, DSC, TGA).

Reviewer 2 Report

Now all my comments have been adressed.

Reviewer 3 Report

The authors have clarified the dubots of my last review. In my opinion the mauscript is acceptable for publication.

This manuscript is a resubmission of an earlier submission. The following is a list of the peer review reports and author responses from that submission.

Round 1

Reviewer 1 Report

This manuscript reported the structurally similar impurity 4-Aminophenol (4-AP) of paracetamol drug can be removed by introducing a certain counterion, namely Salicylic acid (SA), Oxalic acid (OX), L-Tartaric acid (TA), (1S)-(+)-10-Camphorsulfonic acid (CSA) and Hydrochloric acid (HA). Later on, the crystal structure of 4-AP with these counterion salts were solved by single crystal X-ray diffraction. The experimental observations are interesting and meaningful. However, the correlation between impurity removal and the formation of 4-AP salts was not clearly, and at least the experimental evidence is lacking. The stated evidence to support the above correlation is merely from color of the resultant paracetamol products. The UV and HPLC data, nonetheless, does not support the successful detection of impurity of 4-AP or its claimed polymerized substances. The claim to support the products without impurity or additives by the morphology detected from SEM images is not acceptable.

Furthermore, the following concerns are also important to be addressed:

#1. There are some widely acknowledged similarities between salt and co-crystal, and co-crystals have been proposed and employed as a purification method to remove the structurally similar impurity of pharmaceuticals. The authors should mention this fact in the introduction, and clearly pointed out the difference between this study and previous ones.

#2. The quantities of impurities by the color is not directly and evidently. Also, the relationship between the color of products and its real impurity is not clearly. Moreover, the UV and HPLC analyses do not observe the impurity signals, which is not necessary minutely presented and nothing.

#3. Crystal structure analyses were not directly related with the main topic the impurity removal by the formation of 4-AP salts. At least, more elaboration between the two is needed.

#4. In conclusions, the first two sentences have no direct connection with the main findings and evidence support in this study. And in abstract, the elaboration of the main findings is lacking and not clear.

Reviewer 2 Report

The authors describe here a protocol for the purification of paracetamol from aminophenol (AP) by the formation of different aminophenol salts. This topic is of interest to the pharmaceutical industry. The manuscript is focused on the characterization of four new different salts of AP with four acids (oxalic, tartaric, salicilic and camphor sulfonic). The respective crystals have been isolated and investigated by different techniques. The single crystal x-ray structure of each salt have been fully described in the manuscript. A number of other techniques have been reported in the supporting information (IR, DSC, TGA, UV, PXRD) but none or little comments have been done in the manuscript.

In my opinion, it is not clear how these results can be useful for the scope of the manuscript. i.e. the removal of AP from paracetamol. The discussion about the results of the application of the proposed protocol to the purification of paracetamol is missing. The authors reported some HPLC chromatogram in the ESI but no comment on these results have been made in the text. In my opinion some issues need to be discussed.

Is there any quantification of the amount of AP that can be removed with this protocol?  

Is the analytical method enough sensitive to detect the residual AP (up to 50ppm)?

In the spiking experiments, a stoichiometric amount of acid, with respect to the AP, is added to the mixture: it is possible that small amount of the added acid could remain in the final paracetamol sample? Have the authors checked this?

It is not clear how HCl was used; HCl was added as a solution (concentration?) or was it insufflated as a gas? How could be measured the amount of HCl added?

AP impurities can be found as residue from the synthesis of paracetamol or alternatively from paracetamol degradation. Did the authors considered the possibility that stirring paracetamol in the presence of an acid could generate some amount of AP as the result of degradation of paracetamol itself?

My opinion is that the manuscript, in the way it was presented, is essentially a work on the solid state characterization of four new salts of AP. The application of the use of acids for the removal of AP from paracetamol has not been adequately investigated. In my opinion this manuscript would be more appropriate for a specialized journal on solid state characterization. For this reason, at this stage, the manuscript is not suitable for the publication on Molecules.